# Gene redundancy and gene compensation of insulin-like peptides in the oocyte development of bean beetle

Yongqin Li[1], Zheng Fang[1], Leitao Tan[1], Qingshan Wu[1], Qiuping Liu[1], Yeying Wang[1], Qingbei Weng[1,2]*, Qianquan Chen🄳[1]*

**1** School of Life Sciences, Guizhou Normal University, Gui'an, Guizhou, China, **2** Qiannan Normal University for Nationalities, Duyun, Guizhou, China

* wengqingbei@gznu.edu.cn (QW); qqchen@gznu.edu.cn (QC)

**Data Availability Statement:** All relevant data are within the manuscript and its Supporting Information files.

## Abstract

Bean beetle (*Callosobruchus maculatus*) exhibits clear phenotypic plasticity depending on population density; However, the underlying molecular mechanism remains unknown. Compared to low-density individuals, high-density individuals showed a faster terminal oocyte maturity rate. Four insulin-like peptide (ILP) genes were identified in the bean beetle, which had higher expression levels in the head than in the thorax and abdomen. The population density could regulate the expression levels of *CmILP1-3*, *CmILP2-3*, and *CmILP1* as well as *CmILP3* in the head, thorax, and abdomen, respectively. RNA interference results showed that each *CmILP* could regulate terminal oocyte maturity rate, indicating that there was functional redundancy among *CmILPs*. Silencing each *CmILP* could lead to down-regulation of some other *CmILPs*, however, *CmILP3* was up-regulated in the abdomen after silencing *CmILP1* or *CmILP2*. Compared to single gene silencing, silencing *CmILP3* with *CmILP1* or *CmILP2* at the same time led to more serious retardation in oocyte development, suggesting *CmILP3* could be up-regulated to functionally compensate for the down-regulation of *CmILP1* and *CmILP2*. In conclusion, population density-dependent plasticity in terminal oocyte maturity rate of bean beetle was regulated by *CmILPs*, which exhibited gene redundancy and gene compensation.

## Introduction

Insulin/insulin-like growth factor signaling (IIS) pathway works as a nutrition sensor at the systemic level in metazoans [1, 2]. *Bombyxin* is the first identified insect insulin-like peptide (ILP); subsequently, ILPs are extensively identified in invertebrates. For example, eight ILPs are identified in *Drosophila melanogaster* and *Aedes aegypti*, and four ILPs are identified in *Tribolium castaneum* and *Nilaparvata lugens* [3–6]. ILPs exhibit a high degree of structural conservation, whose precursor contains a signal peptide, B-chain, interconnecting C-peptide, and A-chain [1]. After removing the signal peptide and C-peptide, the B-chain and A-chain are linked by disulphide bridges [1]. In *D. melanogaster*, ILPs are expressed mainly in the

**Funding:** This research was funded by National Natural Science Foundation of China (grant number 32060124), Guizhou Normal University (grant number Qianshixinmiao [2021] A11), Guizhou Provincial Science and Technology Foundation (grant number Qiankehejichu[2020]1Y080), the Joint Fund of the National Natural Science Foundation of China and the Karst Science Research Center of Guizhou Province (U1812401), Provincial Program on Platform and Talent Development of the Department of Science and Technology of Guizhou China (grant number [2019]5617, [2019]5655 and [2017]5726-21). The funders had no role in study design, data collection and analysis, decision to publish, or preparation of the manuscript.

**Competing interests:** The authors have declared that no competing interests exist.

nervous system [2]. *Pars intercerebralis* (PI) and *Pars lateralis* (PL) are the two centers of the central neuroendocrine system [2]. PI has a set of median neurosecretory cells (mNSCs) that contain insulin-producing cells (IPCs), the major site for the production of *DmILPs* [2]. At the same time, ILPs are expressed in the midgut, imaginal discs, salivary glands, ovary, and fat body [7–10]. They are involved in many physiological processes, including cell cycle, development, metabolism, feeding behavior, reproduction, stress resistance, diapause, aging, lifespan, and immunity [2, 3, 6, 11–13]. Most of the knowledge about ILPs comes from *D. melanogaster* [2].

Phenotypic plasticity is a benefit for organisms to adapt to changeable environments [14, 15]. For example, the migratory locust (*Locusta migratoria*) exhibits obvious population density-dependent phenotypic plasticity. Compared to low-density locusts, high-density locusts show more active behavior, faster development, and blacker body color [16, 17]. Similarly, under high density, the individuals of *N. lugens* develop into long-winged morphs that can fly a long distance for dispersal, while individuals develop into short-winged morphs that are flightless under low density [18]. Most researches on phenotypic plasticity focus on body color, behavior, and morphology, while few researches study reproduction. The bean beetle (*Callosobruchus maculatus*), an important storage pest, distributes around the world. It can reproduce 11 to 12 generations a year, resulting in up to 100% food loss [19]. It exhibits obvious phenotypic plasticity depending on population density. Compared to low-density individuals, high-density individuals develop more quickly, resulting in a shorter generation time to promote population outbreaks, thereby exacerbating food losses [20]. The effect of population density on terminal oocyte maturity rate that can boost population outbreaks, has not been systematically explored in bean beetle. At the same time, the function of ILPs in bean beetle remains unknown.

In this study, four *CmILPs* were identified from genomic and transcriptomic data of bean beetle. The RNA interference experiments showed that each *CmILP* could regulate the terminal oocyte maturity rate and gene compensation occurred in *CmILPs*.

## Materials and methods

### Insect

Bean beetles were obtained from the wild and identified by morphological and cytochrome-C oxidase subunit 1 (*CO1*) gene sequencing [20]. They were reared in tissue culture glass bottles (0.2 L) with a 5 cm × 5 cm window on the top of the bottle, which was covered with gauze to prevent insects from escaping and to ventilate the bottle. Two pairs of bean beetles were placed in a bottle containing approximately 100 mung beans to establish a low-density experimental population. At the same time, 20 pairs of bean beetles were placed in a bottle containing approximately 100 mung beans to establish a high-density experimental population. Before the experiments, the bean beetles were reared for five generations. The rearing conditions were 28 ± 2° with a 14: 10 light: dark photo regime and 50%-70% humidity.

### Sampling and measurement of terminal oocyte length

Within 12 hours after the emergence of the adults, some females were stored in a -80° refrigerator for subsequent RNA isolation, and the remaining females were soaked in 4% paraformaldehyde solution for 72 hours, subsequently, the ovarioles were separated from each other. The terminal oocytes were captured with a stereomicroscope (SZX7 Olympus, Japan), then the length of terminal oocytes was measured with the software cellSens (Olympus, Japan). Ten terminal oocytes from each female were measured, then the ten terminal oocytes' lengths were

calculated as an average value. The number of females which were used for the measurement of terminal oocyte length would be described in the text.

### Identification of ILPs

The protein sequences of the ILPs of *D. melanogaster*, *T. castaneum*, *N. lugens* and *A. aegypti* were downloaded from NCBI (https://www.ncbi.nlm.nih.gov/) and used as query sequences to determine their orthologous genes from the transcriptome (PRJNA309272) and genome (PRJEB62873) of the bean beetle [21, 22]. Nucleotide sequences from bean beetle were translated into amino acid sequences with Open Reading Frame Finder (https://www.ncbi.nlm.nih.gov/orffinder/). The full length of ILPs' protein sequences of the five species mentioned above were used to construct a phylogenetic tree using Neighbor-Joining algorithms in MEGA 11.0 [23]. Bootstrapped values were calculated with 1000 replications. The substitution model was the Poisson model. Pairwise deletion was used for gaps/missing data treatment.

### Quantitative real-time PCR

Female samples were divided into three parts, i.e., head, thorax, and abdomen. Each part of six individuals was pooled together as one replicate and six biological replicates were performed for each treatment. Total RNA was isolated from the samples using TriQuick Reagent (Solarbio) according to the manufacturer's protocol. The concentration and quality of RNA were determined with Biotek Epoch2. Genomic DNA (gDNA) was removed with DNase in FastKing gDNA Dispelling RT SuperMix (Tiangen). cDNA was reverse-transcribed with two µg of total RNA using the FastKing gDNA Dispelling RT SuperMix. Expression at the mRNA levels were measured using a Talent qPCR PreMix (SYBR Green) (Tiangen) and normalized to ribosome protein 49 [24, 25]. The qRT-PCR conditions were initial denature at 95˚ for 2 min, 40 cycles at 95˚ for 15 s, annealing temperature of 58˚ for 20 s, extension temperature of 72˚ for 20 s, and final extension of 72˚ for 10 min. Melting curve analysis (58–95˚) was applied to all reactions to confirm the specificity of amplification. PCR amplification was performed using the Thermo QuantStudio3. PCR product sequences were validated by Sanger sequencing. Relative gene expression values were calculated using the $2^{-\triangle\triangle Ct}$ method. The primers used for qRT-PCR are presented in S1 Table.

### RNA interference

Four *CmILPs* were amplified using cDNA with gene-specific primers (S1 Table), then cloned into pGEM-T Easy vector (Promega, Madison, USA). Double-stranded RNA (dsRNA) synthesis templates were generated using the vector containing a gene-specific sequence with gene-specific primers containing the T7 promoter sequence at the 5' end of forward primer or reverse primer. dsRNAs targeting the *green fluorescent protein* (dsGFP), *CmILP1* (dsCmILP1), *CmILP2* (dsCmILP2), *CmILP3* (dsCmILP3), and *CmILP4* (dsCmILP4) were synthesized using T7 RiboMAX™ Express Large Scale RNA Production System (Promega). At the pupal stage, approximately one µg (two µg·µl$^{-1}$) of the corresponding dsRNA was injected into each bean beetle abdomen using a nanoliter injector 2000 (World Precision Instruments) with micro glass needle. Three days later, most of the beetles emerged as adults. Within 12 hours after the emergence of the adults, females were collected for silencing efficiency analysis and for measurement of the length of terminal oocytes. The primers used for RNA interference (RNAi) are presented in S1 Table.

## Statistical analysis

Statistical analysis was performed using the IBM SPSS Statistics 26.0 software. Normality and equality of data variances were checked using the Shapiro-Wilk test. For data that showed a normal and homogenous distribution, the student's *t*-test was used for a two-group comparison, while one-way ANOVA was used for the comparison of three groups. Multiple comparisons were conducted with the Student-Newman-Keuls test. For data that did not conform to the normal distribution and (or) equality of variances, comparisons were conducted with the Kruskal-Wallis test. Differences were considered significant if $P < 0.05$. Data are presented as mean ± SEM (standard error of mean).

## Results

### Terminal oocyte maturity rate

Within 12 hours after the emergence of the adults, females were soaked in 4% paraformaldehyde solution for 72 hours. Then the ovarioles were separated from each other to measure terminal oocyte length, which could be used to indicate the oocyte maturity rate [25]. The terminal oocyte length of the high-density bean beetles (H) was significantly longer than that of the low-density bean beetles (L) (Fig 1A and 1B; n = 15 for each group; *t*-test $P < 0.001$). In short, population density could regulate the terminal oocyte maturity rate of bean beetles.

### Identification and expression of ILPs

Four *CmILPs* were identified from the transcriptome (PRJNA309272) and genome (PRJEB62873) of the bean beetle (Fig 2A; S2 Table). The phylogenetic tree showed that *CmILPs*

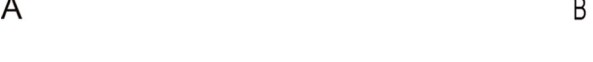

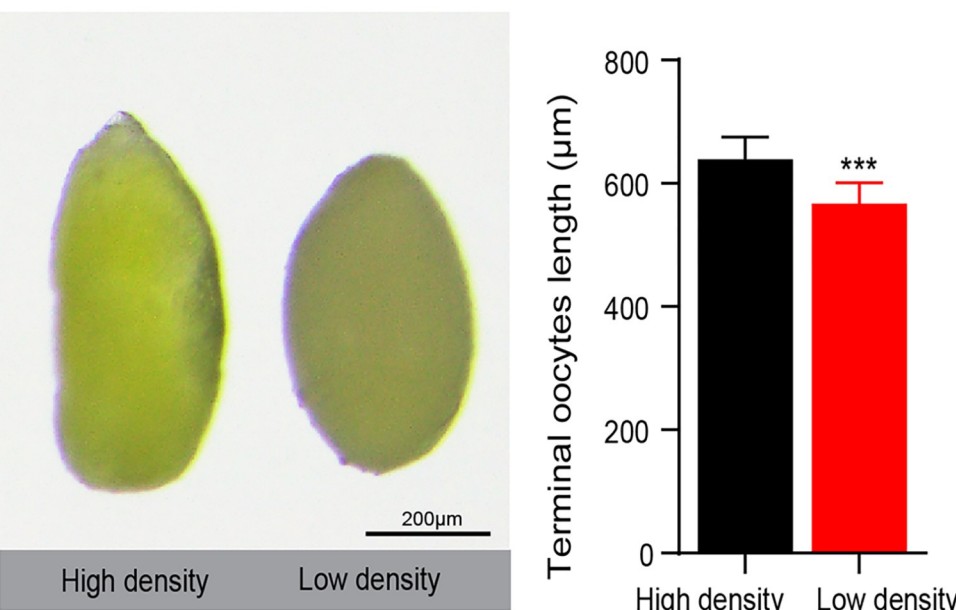

**Fig 1. Effect of population density on terminal oocyte development.** (A) Terminal oocyte morphology. (B) Terminal oocyte length. The error bars represent the standard error of the mean (SEM). Student's *t*-test, \*, $P < 0.05$, \*\*, $P < 0.01$, \*\*\*, $P < 0.001$. The labels are the same as in the other figures.

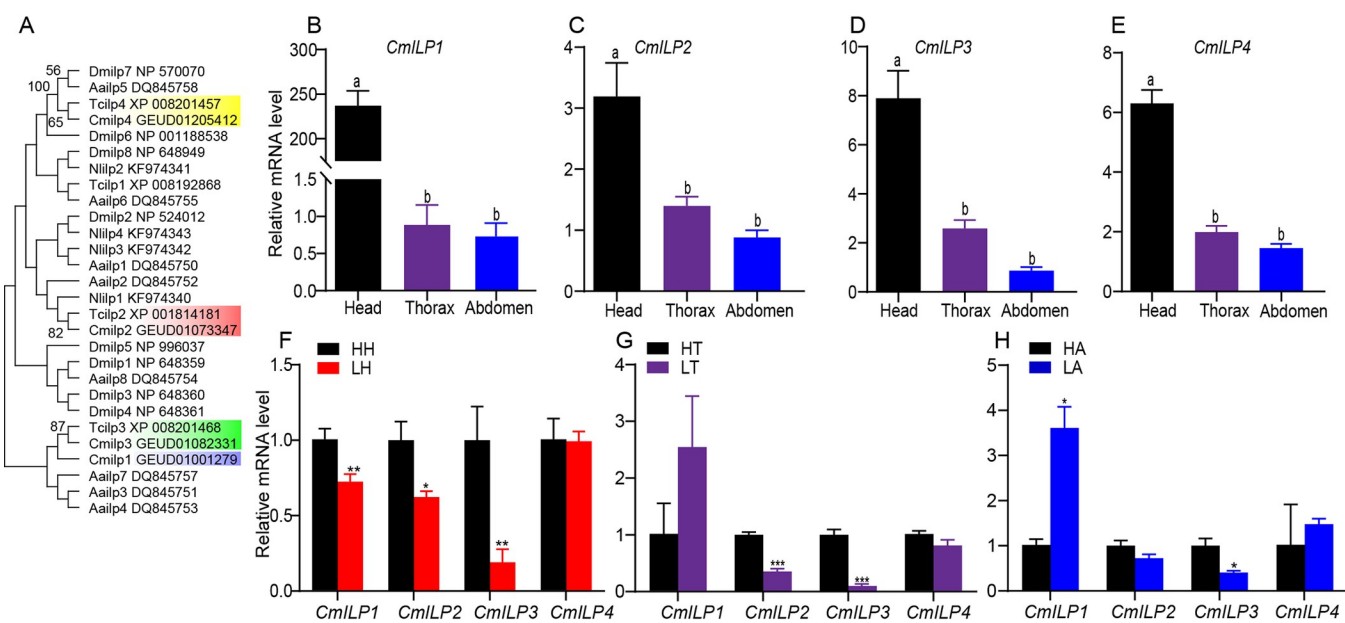

**Fig 2. Expression of insulin-like peptides in *C. maculatus*.** (A) Phylogenetic tree of insulin-like peptides of *C. maculatus* (Cm), *T. castaneum* (Tc), *D. melanogaster* (Dm), *N. lugens* (Nl), and *A. aegypti* (Aa). Bootstrap replicate support percentages are shown next to the branches. (B-E) Tissue-specific expression of *CmILPs*. (F-H) Effect of population density on *CmILPs'* expression. HH (head of high-density individuals) and LH (head of low-density individuals), HT (thorax of high-density individuals) and LT (thorax of low-density individuals), HA (abdomen of high-density individuals) and LA (abdomen of low-density individuals). Entries labeled with different letters indicate statistically significant difference, and those labeled with similar letters indicate statistically non-significant difference.

were clustered with corresponding *TcILPs* except for *CmILP1*, which was clustered with *CmILP3* and *TcILP3*.

The expression level of *CmILP1* in the head was 289.53 times higher than that of the thorax or abdomen (Fig 2B, n = 6 for each group; ANOVA *P* < 0.001). The expression levels of the remaining *CmILPs* in the head were approximately three times higher than in the thorax or abdomen (*CmILP2*, Fig 2C, n = 6 for each group, ANOVA *P* = 0.0010; *CmILP3*, Fig 2D, n = 6 for each group, ANOVA *P* < 0.001; *CmILP4*, Fig 2E, n = 6 for each group, ANOVA *P* < 0.001). There were no differences in the expression levels of *CmILP1*, *CmILP2*, *CmILP3*, and *CmILP4* between the thorax and abdomen.

Compared to H, the cephalic mRNA levels of *CmILP1*, *CmILP2*, and *CmILP3* in L decreased by 28.02%, 37.68%, and 80.96%, respectively (Fig 2F, n = 6 for each group; *t*-test *P* = 0.0060, 0.034, and 0.0020 for *CmILP1*, *CmILP2*, and *CmILP3*, respectively). Compared to H, the thoracic mRNA levels of *CmILP2* and *CmILP3* in L decreased by 64.36% and 90.22%, respectively (Fig 2G, n = 6 and 5 for H and L, respectively; *t*-test *P* < 0.001 for *CmILP2* and *P* = 0.0010 for *CmILP3*). Compared to H, the abdominal *CmILP3* mRNA level in L decreased by 56.48%, however, the *CmILP1* mRNA level in L increased by 3.69-fold (Fig 2H, n = 6 and 5 for H and L, respectively; *t*-test *P* = 0.027 and 0.018 for *CmILP1* and *CmILP3*, respectively). Taken together, the population density could regulate the expression levels of *CmILP1-3*, *CmILP2-3*, and *CmILP1* as well as *CmILP3* in the head, thorax, and abdomen, respectively.

## Function of ILPs in terminal oocyte maturity

To reveal *CmILPs'* function in oocyte development, double-stranded RNAs targeting *CmILPs* were designed. The *CmILP1* mRNA levels were dramatically decreased in the head and abdomen after dsCmILP1 injection (Fig 3A, n = 6, *t*-test *P* < 0.001 for the head; and n = 4, *t*-test

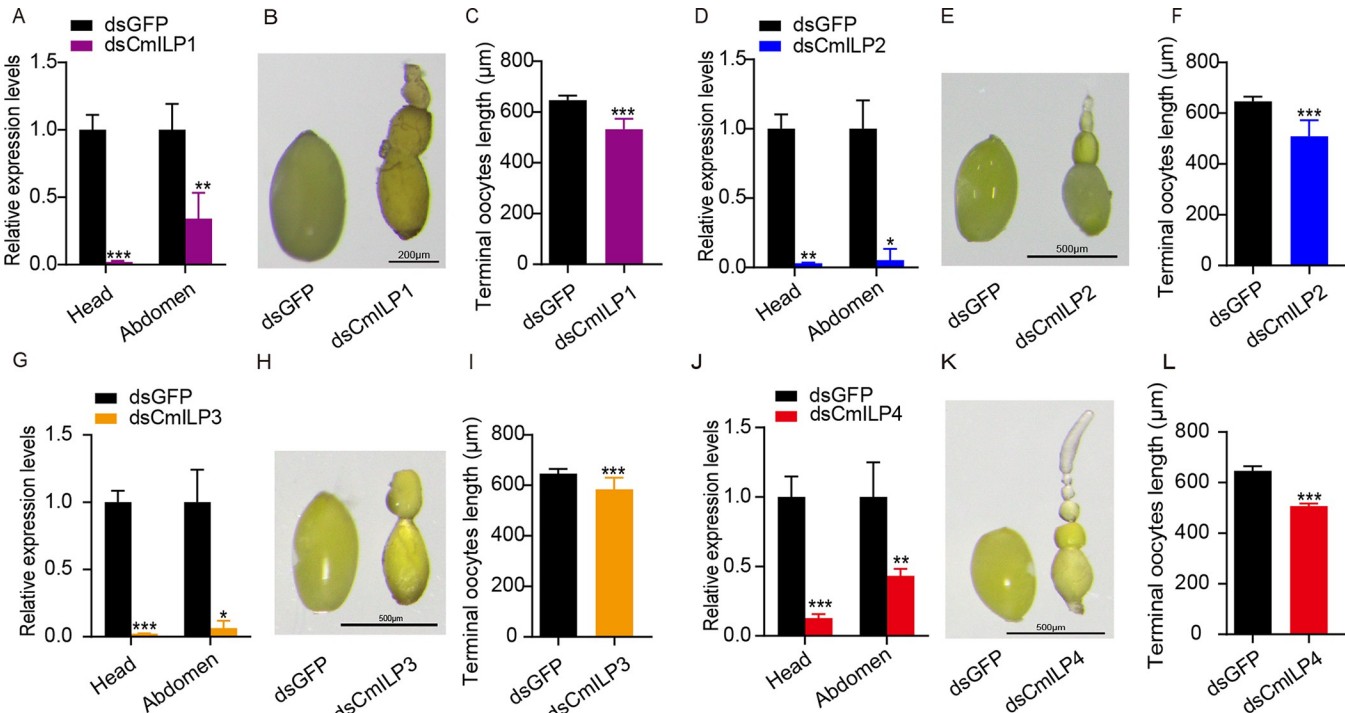

**Fig 3. Effect of silencing each *CmILP* on terminal oocyte length.** (A, D, G, J) Silencing efficiency of *CmILP1-4*. (B, C, E, F, H, I, K, L) Terminal oocyte length after silencing *CmILP1-4*.

*P* = 0.002 for the abdomen). As a result, terminal oocyte length decreased by 21.56% (Fig 3B and 3C, n = 15 for each group; *t*-test *P* < 0.001). Similarly, the mRNA levels of CmILP2, CmILP3 and CmILP4 decreased significantly in the head and abdomen after dsCmILP2, dsCmILP3 and dsCmILP4 injection, respectively (*CmILP2*, Fig 3D, n = 6, *t*-test *P* = 0.0010 for the head and n = 4, *t*-test *P* = 0.018 for the abdomen; *CmILP3*, Fig 3G, n = 6 for each group, *t*-test *P* < 0.001 for the head and *P* = 0.011 for the abdomen; *CmILP4*, Fig 3J, n = 6 for each group, *t*-test *P* < 0.001 for the head and *P* = 0.0050 for the abdomen). Consequently, terminal oocyte length decreased significantly (*CmILP2*, Fig 3E and 3F, n = 15 for each group, *t*-test *P* < 0.001; *CmILP3*, Fig 3H and 3I, n = 15 for each group, *t*-test *P* < 0.001; *CmILP4*, Fig 3K and 3L, n = 15 for each group, *t*-test *P* < 0.001). Taken together, *CmILPs* exhibited redundant roles in regulating oocyte development.

## Inter-regulation of CmILPs

There was no obvious sequence identity at the nucleotide and amino acid level among *CmILPs* (S1 Fig). To unveil the inter-regulation of *CmILPs*, the expression levels of remaining *CmILPs* were determined after silencing each single *CmILP*. After silencing *CmILP1*, the expression levels of *CmILP2* and *CmILP4* decreased in the head (*CmILP2*, Fig 4A, n = 6 for each group; *t*-test *P* = 0.0020; *CmILP4*, Fig 4C, n = 6 for each group; *t*-test *P* = 0.013). However, the mRNA level of *CmILP3* increased 2.38 times in the abdomen (Fig 4B, n = 4 for each group; *t*-test *P* < 0.001; S2 Fig). After silencing *CmILP2*, the mRNA levels of *CmILP1* and *CmILP4* decreased in the head (*CmILP1*, Fig 4D, n = 6 for each group; *t*-test *P* < 0.001; *CmILP4*, Fig 4F, n = 6 for each group; *t*-test *P* = 0.0060), while the mRNA level of *CmILP3* increased by 5.20 times in the abdomen (Fig 4E, n = 4 for each group; *t*-test *P* < 0.001). After suppressing *CmILP3*, the expression levels of *CmILP1*, *CmILP2*, and *CmILP4* decreased in the abdomen

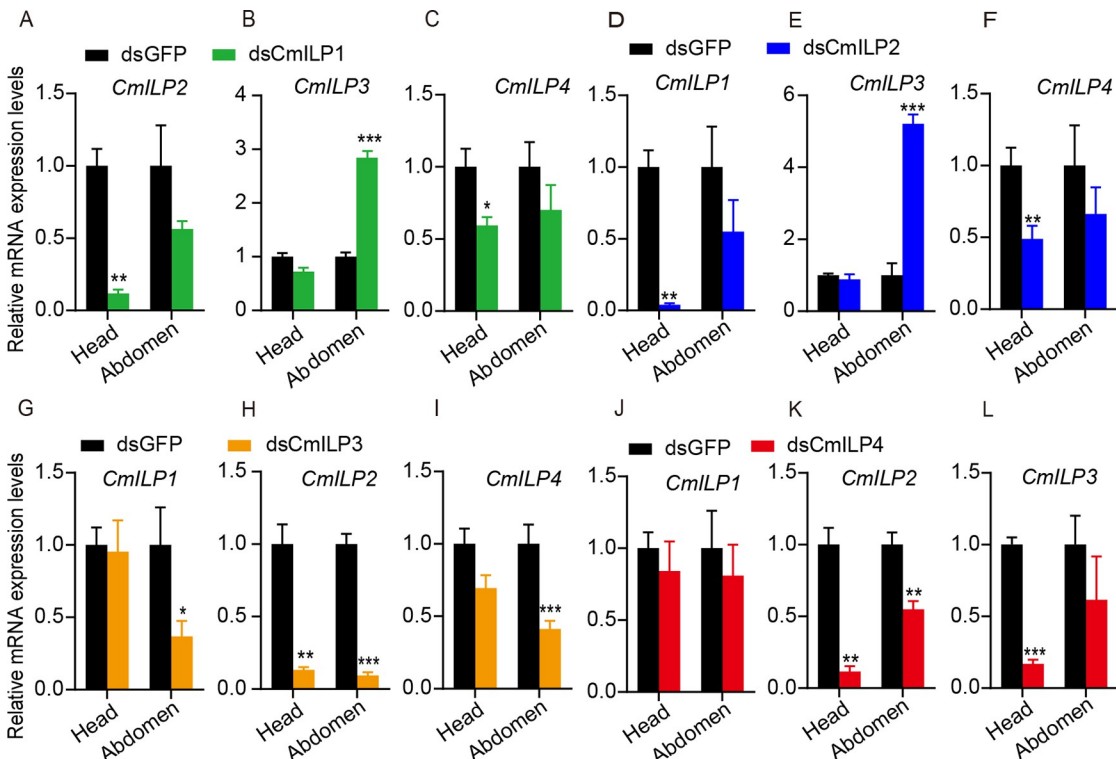

**Fig 4. Regulatory relationship of *CmILPs*.** (A-C) Gene expression levels of *CmILP2*, *CmILP3*, and *CmILP4* after silencing *CmILP1*. (D-F) Gene expression levels of *CmILP1*, *CmILP3*, and *CmILP4* after silencing *CmILP2*. (G-I) Gene expression levels of *CmILP1*, *CmILP2*, and *CmILP4* after silencing *CmILP3*. (J-L) Gene expression levels of *CmILP1*, *CmILP2*, and *CmILP3* after silencing *CmILP4*.

(*CmILP1*, Fig 4G, n = 6 for each group, *t*-test *P* = 0.034; *CmILP2*, Fig 4H, n = 6 for each group, *t*-test *P* < 0.001; *CmILP4*, Fig 4I, n = 6 for each group; *t*-test *P* < 0.001). The expression level of *CmILP2* decreased in the head (Fig 4H, n = 6 for each group; *t*-test *P* = 0.0080). Silencing *CmILP4* resulted in decrease of *CmILP2* mRNA levels in the head and abdomen (Fig 4K, n = 6 for each group; *t*-test *P* = 0.0020 and 0.0050 for the head and abdomen, respectively). Similarly, the *CmILP3* mRNA level decreased by 83.01% in the head (Fig 4L, n = 6 for each group; *t*-test *P* < 0.001). Taken together, there were inter-regulations among *CmILPs*. In particular, *CmILP3* was up-regulated in the abdomen following knockdown of *CmILP1* and *CmILP2*.

## Gene compensation in ILPs

Four *CmILPs* could regulate the oocyte maturity rate in bean beetle. Owing to the mRNA level of *CmILP3* increased in the abdomen after silencing *CmILP1* or *CmILP2*, dsCmILP3 with dsCmILP1 or dsCmILP2 were injected into bean beetles at the same time. After co-injection of dsCmILP1 and dsCmILP3, the mRNA levels of *CmILP1* and *CmILP3* decreased significantly in the head and abdomen (*CmILP1*, Fig 5A, n = 6, *t*-test *P* < 0.001 for the head and n = 4, *t*-test *P* = 0.0010 for the abdomen; *CmILP3*, Fig 5B, n = 6, *t*-test *P* < 0.001 for the head and n = 4, *t*-test *P* = 0.042 for the abdomen). As a result, the ovaries of 62.96% dsCmILP1 & dsCmILP3 injection females were not developed enough to separate terminal oocytes (Fig 5C). The remaining individuals' ovaries were developed enough to separate the terminal oocytes. However, the terminal oocyte length decreased by 35.20% (Fig 5D and 5E, n = 12 and 10 for dsGFP and dsCmILP1&ILP3, respectively; *t*-test *P* < 0.001). At the same time, the mRNA

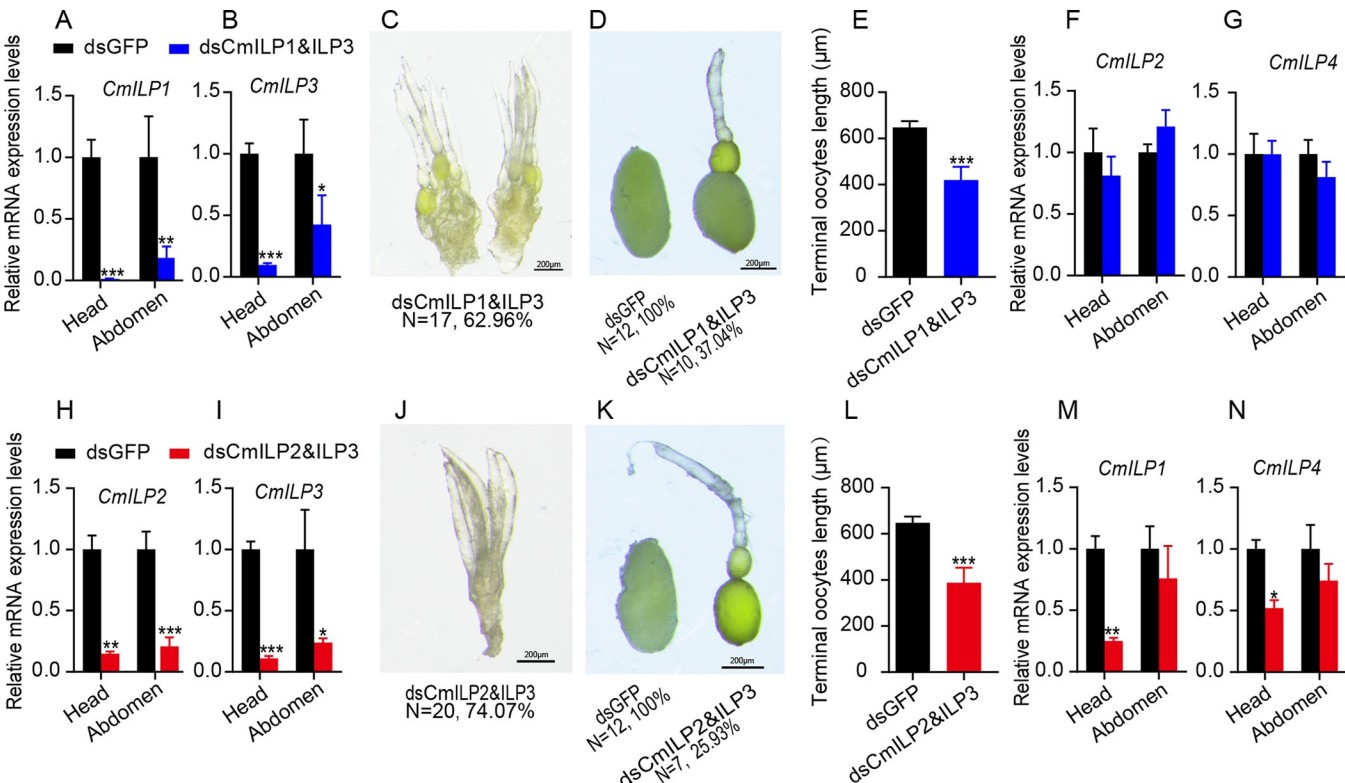

**Fig 5. Redundancy and compensation of *CmILPs* in terminal oocyte maturity rate.** (A, B) Silencing efficiency of *CmILP1* and *CmILP3*. (C) Ovary development status, (D) terminal oocyte, (E) terminal oocyte length after silencing *CmILP1* and *CmILP3*. Expression levels of *CmILP2* (F) and *CmILP4* (G) after silencing *CmILP1* and *CmILP3*. (H, I) Silencing efficiency of *CmILP2* and *CmILP3*. (J) Ovary development status, (K) terminal oocyte, (L) terminal oocyte length after silencing *CmILP2* and *CmILP3*. Expression levels of *CmILP1* (M) and *CmILP4* (N) after silencing of *CmILP2* and *CmILP3*.

levels of *CmILP2* and *CmILP4* did not change in the head and abdomen (*CmILP2*, Fig 5F, n = 6 for each group, *t*-test *P* = 0.51 and 0.19 for the head and abdomen, respectively; *CmILP4*, Fig 5G, n = 6 for each group, *t*-test *P* = 0.99 and 0.31 for the head and abdomen, respectively).

After co-injection of dsCmILP2 and dsCmILP3, the mRNA levels of *CmILP2* and *CmILP3* decreased by approximately 80% in the head and abdomen (*CmILP2*, Fig 5H, n = 6 for each group, *t*-test *P* = 0.0020 for the head and *P* < 0.001 for the abdomen; *CmILP3*, Fig 5I, n = 6 for each group, *t*-test *P* < 0.001 for the head and *P* = 0.025 for the abdomen). The ovaries of all dsGFP injection females were fully developed; however, the ovaries of 74.07% of dsCmILP2 & dsCmILP3 injection females were not developed enough to separate terminal oocytes (Fig 5J). The remaining individuals' ovaries were developed sufficiently to separate the terminal oocytes, however, the terminal oocyte length decreased by 40.19% (Fig 5K and 5L, n = 12 and 7 for dsGFP and dsCmILP2 & dsCmILP3, respectively; *t*-test *P* < 0.001). At the same time, the mRNA levels of *CmILP*1 and *CmILP4* decreased in the head (*CmILP*1, Fig 5M, n = 6 for each group, *t*-test *P* = 0.0010; *CmILP4*, Fig 5N, n = 6 for each group; *t*-test *P* = 0.020). Taken together, *CmILP3* could functionally compensate for the down-regulation of *CmILP1* and *CmILP2*.

## Discussion

Compared to low-density bean beetles, high-density individuals had a faster terminal oocyte maturity rate, suggesting that bean beetles exhibited phenotypic plasticity in reproduction

depending on population density. Under high density, individuals who had a faster maturity rate could mate and lay eggs earlier. As a result, their offsprings might have advantages in development, competition for food, and finding mates. Bean beetles laid eggs on seed and hatched as larvae that penetrated seed coat for development [20]. Earlier larvae might have an advantage in competition for food and space. Similar phenomenon occurs in migratory locusts [25]. In locusts, cannibalism for supplementing nutrition, occurs with the increase of population density [26]. Early developmental individuals might have an advantage in killing and consuming conspecific individuals. Taken together, population density-dependent phenotypic plasticity in oocyte maturity rate is a benefit for organisms to adapt to changeable environments [14, 15].

Four *CmILPs* were identified from the transcriptome and genome of bean beetle (Fig 2A). The phylogenetic tree showed that *CmILPs* were clustered with corresponding *TcILPs* except for *CmILP1*, which was clustered with *CmILP3* and *TcILP3*. Owing to ILPs exhibited a high degree of structural conservation rather than amino acid sequence conservation [1], the phylogenetic tree was constructed using amino acid sequence. As a result, it was reasonable that the numbers did not coincide among different species.

The expression levels of *CmILPs* in the head were approximately 3~5-fold higher compared to the thorax or abdomen, however, the expression level of *CmILP1* in the head was 289.53 times higher as that in the thorax or abdomen. In *D. melanogaster*, *ILPs* are expressed mainly in IPCs of the nervous system [2]. For example, *DmILP1*, *DmILP2*, *DmILP3*, and *DmILP5* are expressed mainly in IPCs [2]. Meanwhile, *DmILPs* are widely expressed in other tissues, including the midgut, imaginal discs, salivary glands, ovary, and fat body [7, 8]. Similarly, all *ILPs* are expressed in the fat body of *T. castaneum*; three *ILPs* (*AaILP2*, *AaILP5*, and *AaILP6*) are expressed in the midgut, thoracic wall, and abdomen wall of *A. aegypti* [9, 10]. Compared to low-density individuals, high-density individuals had higher expression levels of *CmILP1*-3 in the head. The expression levels of *CmILP2* and *CmILP3* in the thorax of high-density individuals were higher than that of low-density individuals. For the abdomen, *CmILP3* was expressed higher in high-density individuals than in low-density individuals, however, *CmILP1* was expressed higher in low-density individuals than in high-density individuals. With the increase of population density, the seed competition for oviposition among bean beetles would be enhanced. Taken together, *ILPs* are widely expressed in several tissues, and population density could regulate *ILPs'* expression.

RNAi experiments showed that each *CmILP* could regulate the oocyte maturity rate in bean beetles, suggesting there was functional redundancy among *CmILPs*. In red flour beetle, *ILPs* activate the phosphatidylinositol-3-kinase (PI3K) signaling pathway by binding the insulin receptor (InR), and then forkhead box O (FOXO) is phosphorylated [27]. After phosphorylation, FOXO cannot bind to the promoter of *vitellogenin* (*Vg*) to act as a repressor [9]. Then, *Vg* is expressed in the fat body, and reproduction begins. ILPs are ligands for the insulin/insulin-like growth factor signaling (IIS) pathway, which works as a sensor of nutrition at the systemic level [1]. The IIS pathway, a conservative pathway in all metazoans [2], is a response for multicellular animals to adapt their costly activities according to the environment [3].

To reveal the inter-regulation of *CmILPs*, all *CmILPs'* expression levels were determined after silencing each *CmILP*. The mRNA levels of *CmILP2* and *CmILP4* decreased in the head, however, the mRNA level of *CmILP3* was up-regulated in the abdomen after silencing *CmILP1*. The mRNA levels of *CmILP1* and *CmILP4* decreased in the head, however, the mRNA level of *CmILP3* was up-regulated in the abdomen after silencing *CmILP2*. The mRNA levels of *CmILP1* and *CmILP4* decreased in the abdomen, and the mRNA levels of *CmILP2* decreased in the head and abdomen after silencing *CmILP3*. The mRNA levels of *CmILP2* decreased in the head and abdomen, and the mRNA level of *CmILP3* decreased in the head

after silencing *CmILP4*. In *Drosophila*, when *DmILP2* was mutant, the mRNA levels of *DmILP3* and *DmILP5* were up-regulated by 1.8 and 1.9-fold, respectively. When *DmILP5* was mutant, the mRNA level of *DmILP3* increased by 1.7-fold. When *DmILP3* was mutant, the mRNA levels of *DmILP2* and *DmILP5* decreased by 67%, which indicated that *DmILP3* could regulate the expression of *DmILP2* and *DmILP5* [3]. When the mRNA levels of *DmILP2*, *DmILP3*, and *DmILP5* decreased in the brain, the mRNA level of *DmILP6* was up-regulated in the fat body [3], suggesting that there was negative feedback in the expression of *DmILPs* between the MNCs in the central nervous system and the fat body in peripheral tissue [3]. Similarly, knockdown of *BgILP5* in *Blattella germanica*, the mRNA level of *BgILP3* increases by 1.8-fold [28]. In migratory brown planthopper, the expression level of *NlILP2* decreased when the knockdown of *NlILP1*, *NlILP3*, or *NlILP4*; the expression level of *NlILP1* decreased when the knockdown of *NlILP2* [6]. In short, there were inter-regulations between *ILPs*; however, the mechanism remained unknown.

*CmILP3* was up-regulated in the abdomen after silencing *CmILP1* and *CmILP2*, which indicated that *CmILP3* showed a functional compensation for the knockdown of *CmILP1* and *CmILP2*. When *CmILP2* and *CmILP3* were silenced at the same time, the mRNA levels of *CmILP1* and *CmILP4* decreased in the head, and the oocyte maturity rate was more seriously decreased than silencing a single *CmILP*. When *CmILP3* and *CmILP1* were silenced at the same time, the terminal oocyte maturity rate was more seriously decreased than silencing a single *CmILP*. It seemed that *CmILP3* could functionally compensate for the down-regulation of *CmILP1* and *CmILP2*. In *Drosophila*, females with mutations in *DmILP2*, *DmILP3*, *DmILP5*, and *DmILP6*, have 25%, 22%, 18%, and 46% decrease in egg production, respectively [3]. When mutations occur in *DmILP2* and *DmILP3* at the same time, the egg production decreases by 27% [3]. More seriously, the egg production decreases by 69% when mutations occur in *DmILP2*, *DmILP3*, and *DmILP5* at the same time [3]. These data suggest that *DmILP2*, *DmILP3*, and *DmILP5* have redundant roles in the regulation of egg production [3]. In red flour beetle, *TcILP1*, *TcILP2*, and *TcILP3* are involved in nymphal development [6]. Compared to single gene knockdown, the combination of two genes of *NlILP1*, *NlILP2*, and *NlILP3* knockdowns has a more serious delay in nymphal development [6]. When *NlILP1*, *NlILP2*, and *NlILP3* are silenced at the same time, the strongest delay in nymphal development occurs [6]. Gene compensation is widely present in animals and plants. The *egfl7* mutant does not exhibit an obvious phenotype in zebrafish. Further experiments show that *emilin*, which has similar function and sequence homology to *egfl7*, is up-regulated to compensate for the loss of *egfl7* [29]. Similarly, after knockout of *Actg1* in mouse embryonic stem cells, the mRNA level of *Actg2* increases [30].

Previous studies show that transcripts with premature termination codons are degraded through the non-sense-mediated decay (NMD) pathway, and homologous genes can be up-regulated for function compensation [29]. Sequence similarity is required for the activation of gene compensation [30]. However, the expression level of endogenous *nid1a* does not change significantly in transgenic zebrafish, which harbors premature termination codons in the *nid1a* transgene, suggesting that there are differences in gene compensation among gene families [31]. There is no obvious sequence similarity among *ILPs* (S2 Table), and the mechanism of *ILPs'* functional compensation needs further study.

## Conclusions

Population density could regulate the terminal oocyte maturity rate of bean beetles and *CmILPs'* expression. Each *CmILP* could regulate terminal oocyte maturity rate, suggesting that there was functional redundancy among *CmILPs*. There were inter-regulations among

*CmILPs. CmILP3* was up-regulated to compensate for the down-regulation of *CmILP1* and *CmILP2.*

## Supporting information

**S1 Fig. Sequence alignment of *CmILPs*.** (A) Protein sequence, (B) nucleotide sequence.
(TIF)

**S2 Fig.** Expression of insulin-like peptides in head (A), thorax (B), and abdomen (C).
(TIF)

**S1 Table. Primers for RT-qPCR and RNAi.**
(DOCX)

**S2 Table. Sequence identity and similarity among ILPs.**
(DOCX)

**S3 Table. Raw data.**
(XLSX)

## Author Contributions

**Conceptualization:** Qianquan Chen.

**Formal analysis:** Yongqin Li, Zheng Fang, Leitao Tan, Qingshan Wu, Qiuping Liu, Yeying Wang, Qianquan Chen.

**Funding acquisition:** Yeying Wang, Qingbei Weng, Qianquan Chen.

**Investigation:** Yongqin Li, Qianquan Chen.

**Writing – original draft:** Yongqin Li, Qianquan Chen.

**Writing – review & editing:** Qingbei Weng, Qianquan Chen.

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
