## [Decision Letter · Decision Letter 0]

2 Jan 2024

PONE-D-23-41217Gene redundancy and gene compensation of insulin-like peptides in the sexual maturity rate of bean beetlePLOS ONE

Dear Dr. Chen,

Thank you for submitting your manuscript to PLOS ONE. After careful consideration, we feel that it has merit but does not fully meet PLOS ONE’s publication criteria as it currently stands. Therefore, we invite you to submit a revised version of the manuscript that addresses the points raised during the review process. Please submit your revised manuscript by Feb 16 2024 11:59PM. If you will need more time than this to complete your revisions, please reply to this message or contact the journal office at plosone@plos.org. Please include the following items when submitting your revised manuscript:A rebuttal letter that responds to each point raised by the academic editor and reviewer(s). You should upload this letter as a separate file labeled 'Response to Reviewers'.A marked-up copy of your manuscript that highlights changes made to the original version. You should upload this as a separate file labeled 'Revised Manuscript with Track Changes'.An unmarked version of your revised paper without tracked changes. You should upload this as a separate file labeled 'Manuscript'.

We look forward to receiving your revised manuscript.

Kind regards,

J Joe Hull, Ph.D.

Academic Editor

PLOS ONE

Journal Requirements:

   "This research was funded by National Natural Science Foundation of China (grant number 32060124), Guizhou Normal University (grant number Qianshixinmiao [2021] A11), Guizhou Provincial Science and Technology Foundation (grant number Qiankehejichu[2020]1Y080), the Joint Fund of the National Natural Science Foundation of China and the Karst Science Research Center of Guizhou Province (U1812401), Provincial Program on Platform and Talent Development of the Department of Science and Technology of Guizhou China (grant number [2019]5617, [2019]5655 and [2017]5726-21)."

Additional Editor Comments:

In addition to the Reviewer comments (appended at the bottom of this email), I have provided a number of my own comments (see below). 1) I highly encourage you to seek the input from a scientific editing service or a native English-speaking colleague with molecular experience.

2) For the Introduction, shift the focus of the first paragraph away from mouse & zebrafish to insects

3) Results - if data is presented in a figure, then there is no need to also include it in the text. As currently written, the readability of the Results is hampered by the numerous values listed. Consider highlighting the main points in the Figures rather than listing every percentage value.

4) Provide a sequence alignment figure showing the lack of identity mentioned in line 205.

5) Provide significantly more methodological details. As currently written, they are insufficient to allow for replication.

- provide accession/bioproject numbers for the transcriptome and genome searched. Also, provide accession numbers for the sequences used as queries.

- In the phylogenetic analysis, provide information/details on bootstrap support, the model/method used, how gaps/missing data were treated, and indicate if full-length sequences were used. Was the signal peptide removed prior to analysis? Also, provide accession numbers for the sequences used to construct the phylogenetic tree. 

- Was gDNA removed prior to cDNA synthesis?

- Is there a reference indicating the suitability of using rpl49 as a reference gene?

- Were the PCR product sequences validated?

- What were the qRT-PCR conditions?

- How was the dsRNA synthesis template generated?

- How were dsRNAs injected? was a microinjection system used? where was the injection site? what types of needles were used?

- By "silencing efficiency" do the authors mean confirmation of transcript knockdown? provide PCR conditions

- How were terminal oocyte measurements made?

6) In the Results, indicate how the ILPs were identified (ie based on sequence homology with query sequences or via previous annotation) and indicate what the sequence identity/similarity was with the other ILPs.

Reviewers' comments:

Reviewer's Responses to Questions

**Comments to the Author**

1. Is the manuscript technically sound, and do the data support the conclusions?

Reviewer #1: Yes

Reviewer #2: Yes

2. Has the statistical analysis been performed appropriately and rigorously? 

Reviewer #1: No

Reviewer #2: Yes

3. Have the authors made all data underlying the findings in their manuscript fully available?

Reviewer #1: No

Reviewer #2: Yes

4. Is the manuscript presented in an intelligible fashion and written in standard English?

Reviewer #1: No

Reviewer #2: Yes

5. Review Comments to the Author

Reviewer #1: The manuscript by Li and co-workers reports the occurrence of four ILPs in the bean beetle, their expression in different culture conditions and parts of the body. In addition, they report the effects of different RNAi treatments on oocyte length and ILP mRNA levels. The work is generally well executed and the results are interesting. However, they are quite scarce. Only the expression of ILPs is quantified and no others analyses or experiments are done beyond measuring the length of the oocytes.

In addition, some other questions must be considered.

-Data availability: According to the PLoS One data availability policies, the values behind the means, standard deviations and other measures must be available in the manuscript or in a repository. This provision is not met in the case of this work.

-The manuscript needs a revision of the English language of the text.

-In Material and methods, it is stated that qPCR calculations are performed with the 2-ΔΔCt method, but in the case of Fig.2 B-D it seems that 2-ΔCt instead of 2-ΔΔCt has been used. Is this right?

-Material and methods, statistical analysis: Levene test does not analyze the normality of data. If you want to analyze normality you must use an ad hoc test as, for example, Shapiro-Wilk.

-The results are overexplained and reading the text becomes not only tedious, but even complicated.

-Line 73. Please change “maculate” by “maculatus”.

-Results sexual maturity rate. Although it is mentioned in the Material and Methods section, please indicate also here at what time did you measure the terminal oocyte length for both high- and low-density animals.

-I think it is necessary for the authors to include the sequences of the bean beetle ILPs in NCBI or in another public repository, at least from the time of publication, but I have not been able to find any Accession Number or reference in this regard.

-Discussion: Phylogenetic tree. The number of each ILP usually does not correspond to that determined by the orthology with other ILPs, since it is very difficult to find the orthologous peptides (see, for example, Antonova et al 2012. Insulin-Like Peptides: Structure, Signaling, and Function. In: Gilbert, L. I. (Ed.), Insect Endocrinology. Elsevier, pp. 63–92). In the case of Callosobruchus, I understand that the ILPs have been named because of their similarity to those of Tribolium, a relatively close species. However, that will not be the case with the other species included in the phylogenetic tree, so it is normal that the numbers do not coincide. I consider the phylogenetic tree to be of limited interest.

-There is a clear example of compensation at the level of ILP expression and not only at reproduction or development level in the case of the cockroach Blatella germanica (Castro-Arnau et al. 2019. J. Insect Physiol. 114, 57-67).

-Discussion, lines 361-362. Reference 10 refers to Nilaparvata and not to Tribolium.

-Discussion, lines 367-369. Please, indicate the organism to which this statement refers.

-How do you explain that dsCmILP3 alone and dsCmILP4 alone produce 86.80% and 88.17% reduction of brain CmILP2 mRNA levels, respectively, but dsCmILP3&ILP4 do not modify them?

-First paragraph of the Introduction and last paragraph of the discussion. In the dsRNA treatments, the RNA is degraded through the RNAi pathway using the RNase activity of Argonaute in the RNA-induced silencing complex (RISC), and not using the nonsense-mediated decay (NMD) pathway. Then, I don’t think the information showed in those paragraphs is really relevant.

-Figure 2A. What do the colors mean?

Reviewer #2: In the manuscript by Li et al., the authors identified the expression pattern of insulin like peptides in thebean beetle, and examined the function of these ILPs in oocyte development by RNAi. The experiments are well designed, and the results are solid and support the main conclusions. The manuscript is presented in an intelligible fashion and well written. I only have a few concerns that the authors should address:

1. The authors stated that the female sexual maturity rate could be indicated by the length of terminal oocytes. I feel that this should be better explained in the Introduction. Why and how does long terminal oocyte represent fast sexual maturity rate? In my opion, sexual maturation means the females are ready for mating with males and laying eggs. A group of female beetles have longer terminal oocytes at 12hrs after emergence does not necessarily mean that they will mature faster. If no further experimental evidence could be provided, I would suggest to describe the results as ILPs are involved in oocyte development or terminal oocyte maturation.

2. The first paragraph of the introduction focuses on "Gene redundancy", which seems not to be important and necessary. The examples in this paragraph are not very relevant with ILPs nor insects. Gene redundancy and compensation have been studied in Drosophila ILPs, as the authors described in Discussion.

3. The writtings could be further improved.

6. PLOS authors have the option to publish the peer review history of their article (what does this mean?). If published, this will include your full peer review and any attached files.

Reviewer #1: No

Reviewer #2: No

---

## [Author Response · Author response to Decision Letter 0]

16 Feb 2024

We have carefully read the comments from the editor and reviewers. The comments and suggestions are very valuable for us to improve this manuscript. 

With the modifications, we think that the revised manuscript has been substantially improved in terms of data presentation and analyses, and appreciate your consideration for its publication in PLOS ONE.

---

## [Decision Letter · Decision Letter 1]

29 Feb 2024

PONE-D-23-41217R1Gene redundancy and gene compensation of insulin-like peptides in the oocyte development of bean beetlePLOS ONE

Dear Dr. Chen,

Thank you for submitting your revised manuscript to PLOS ONE. After careful consideration, we feel that it has merit but does not fully meet PLOS ONE’s publication criteria as it currently stands. Therefore, we invite you to submit a revised version of the manuscript that addresses the points raised during the review process.  Reviewer 1 had a number of additional suggestions for you to consider that would further enhance the overall clarity of the manuscript. In addition, I would recommend that further copy editing via a native English speaking colleague or editing service be considered. Although many grammatical problems in the original manuscript were addressed, some issues still persist.  

We look forward to receiving your revised manuscript.

Kind regards,

J Joe Hull, Ph.D.

Academic Editor

PLOS ONE

Journal Requirements:

Reviewers' comments:

Reviewer's Responses to Questions

**Comments to the Author**

1. If the authors have adequately addressed your comments raised in a previous round of review and you feel that this manuscript is now acceptable for publication, you may indicate that here to bypass the “Comments to the Author” section, enter your conflict of interest statement in the “Confidential to Editor” section, and submit your "Accept" recommendation.

Reviewer #1: (No Response)

Reviewer #2: All comments have been addressed

2. Is the manuscript technically sound, and do the data support the conclusions?

Reviewer #1: Yes

Reviewer #2: Yes

3. Has the statistical analysis been performed appropriately and rigorously? 

Reviewer #1: Yes

Reviewer #2: Yes

4. Have the authors made all data underlying the findings in their manuscript fully available?

Reviewer #1: Yes

Reviewer #2: Yes

5. Is the manuscript presented in an intelligible fashion and written in standard English?

Reviewer #1: Yes

Reviewer #2: Yes

6. Review Comments to the Author

Reviewer #1: The new version of the manuscript has incorporated most of the changes suggested by this reviewer. However, I still have some comments regarding the new version. I indicate these comments below.

-There are some parts of the first paragraph of the introduction that are repeated in the discussion. I think the entire first paragraph of the introduction should be removed and all information passed through, just once, into the discussion.

-Given the bootstrap numbers observed in the phylogenetic tree, is it possible that the Bootstrapped values were calculated with 100 replications instead of 1000 replications as stated on line 110?

-Please, indicate how did you eliminate genomic DNA.

-Perhaps due to the new tree that positions the ILPs in places different from those of the original manuscript, there must have been an exchange of names between CmILP1 and CmILP4. Make sure this nomenclature change has also been corrected in the raw data. I think that, at least for some values in Fig.2, that change has not been made.

- I would like to see, even if it is in supplementary material, a figure that directly compares the expressions of the different ILPs in the different tissues (head, thorax and abdomen). That is: a graph with the four ILPs in the head, another in the thorax and another in the abdomen. I think this result is interesting to understand to what extent the increase in ILP3 expression in the abdomen of dsILP1 and dsILP2 can compensate for the reduction in ILP1 and ILP2, respectively. The fact that in Figure 2 B-E the expression in the abdomen for each of the ILPs is used as a “calibrator” makes the comparison of the ILPs in each tissue not direct because it depends on the respective expressions in the abdomen.

-The text in the section “Function of ILPs in terminal oocyte maturity rate” could be greatly reduced considering that the figures are already there.

-How many different animals were used for terminal oocyte length measurements and how many oocytes were measured from each animal? Please, indicate this somewhere in the manuscript.

Reviewer #2: The authors have addresed my concerns as well as those risen by another reviewer and the editor. I have no further questions.

7. PLOS authors have the option to publish the peer review history of their article (what does this mean?). If published, this will include your full peer review and any attached files.

Reviewer #1: No

Reviewer #2: No

---

## [Author Response · Author response to Decision Letter 1]

12 Apr 2024

we revised manuscript according to the comments.

---

## [Editor Report · Decision Letter 2]

17 Apr 2024

Gene redundancy and gene compensation of insulin-like peptides in the oocyte development of bean beetle

PONE-D-23-41217R2

Dear Dr. Chen,

We’re pleased to inform you that your manuscript has been judged scientifically suitable for publication and will be formally accepted for publication once it meets all outstanding technical requirements.

Kind regards,

J Joe Hull, Ph.D.

Academic Editor

PLOS ONE